# Relating Dry Friction to Interdigitation of Surface Passivation Species: A Molecular Dynamics Study on Amorphous Carbon

**DOI:** 10.3390/ma15093247

**Published:** 2022-04-30

**Authors:** Kerstin Falk, Thomas Reichenbach, Konstantinos Gkagkas, Michael Moseler, Gianpietro Moras

**Affiliations:** 1Fraunhofer IWM, MicroTribology Center μTC, Woehlerstr. 11, 79108 Freiburg, Germany; thomas.reichenbach@iwm.fraunhofer.de (T.R.); michael.moseler@iwm.fraunhofer.de (M.M.); gianpietro.moras@iwm.fraunhofer.de (G.M.); 2Institute of Physics, University of Freiburg, Hermann-Herder-Str. 3, 79104 Freiburg, Germany; 3Material Engineering Division, Toyota Motor Europe NV/SA, Technical Center, Hoge Wei 33B, B-1930 Zaventem, Belgium; konstantinos.gkagkas@toyota-europe.com

**Keywords:** boundary lubrication, surface passivation, dry friction, molecular dynamics, tribology, diamond-like carbon

## Abstract

Friction in boundary lubrication is strongly influenced by the atomic structure of the sliding surfaces. In this work, friction between dry amorphous carbon (a-C) surfaces with chemisorbed fragments of lubricant molecules is investigated employing molecular dynamic simulations. The influence of length, grafting density and polarity of the fragments on the shear stress is studied for linear alkanes and alcohols. We find that the shear stress of chain-passivated a-C surfaces is independent of the a-C density. Among all considered chain-passivated systems, those with a high density of chains of equal length exhibit the lowest shear stress. However, shear stress in chain-passivated a-C is consistently higher than in a-C surfaces with atomic passivation. Finally, surface passivation species with OH head groups generally lead to higher friction than their non-polar analogs. Beyond these qualitative trends, the shear stress behavior for all atomic- and chain-passivated, non-polar systems can be explained semi-quantitatively by steric interactions between the two surfaces that cause resistance to the sliding motion. For polar passivation species electrostatic interactions play an additional role. A corresponding descriptor that properly captures the interlocking of the two surfaces along the sliding direction is developed based on the maximum overlap between atoms of the two contacting surfaces.

## 1. Introduction

Reducing friction under boundary lubrication conditions plays an increasingly important role in the development of mechanical systems with high energy efficiency. Boundary lubrication is not only relevant for sliding systems that operate under dry conditions, but also for lubricated tribological systems operating at low speeds or with low-viscosity lubricants, which are used to lower friction in the hydrodynamic regime [1]. In this context, hard, hydrogen-free carbon coatings, such as diamond and tetrahedral amorphous carbon (ta-C), are emerging as high-performance tribological materials, as evidenced by several reports of ultra- and super-low friction obtained in boundary lubrication with water [2,3], alcohols [4] or organic friction modifiers [5,6,7]. 

As both diamond and ta-C are likely to develop a thin amorphous carbon (a-C) surface layer with relatively low density in tribological contacts under boundary lubrication conditions [8,9,10,11,12], the tribochemical mechanisms leading to superlow friction in diamond and ta-C coatings are often analogous. The tribochemical decomposition of the lubricant molecules at the contacts between surface asperities produces chemical species (e.g., hydrogen and oxygen) that chemisorb on the two surfaces and chemically passivate them [7,11,13,14,15]. Surface passivation can occur by chemical saturation of surface dangling bonds with H- and O-related species (e.g., H atoms, OH groups) or by surface aromatic structures that can form during plastic shear deformation of cold-welded, O- and H-doped a-C interfaces [7,16]. In both cases, the result is an almost atomically flat surface with a very low corrugation of the contact potential energy surface (CPES), which is a prerequisite for superlow friction [17,18,19].

Atomistic simulations can be used to relate surface chemical passivation and the resulting CPES to the interface shear stress induced by the relative motion of two surfaces in a dry tribological contact. Owing to system-size limitations, the macroscopic surface roughness cannot be explicitly described by atomistic simulations and the measured shear stress is often not directly comparable to the friction measured in experiments. Nevertheless, these studies are useful to understand how surface chemistry locally affects the shear resistance of sliding interfaces at asperity-asperity contacts. A correlation between the shear stress of a dry tribological interface and the corrugation of the CPES was shown using density functional theory (DFT) simulations for H- and OH-terminated diamond surfaces [3] and it was directly demonstrated by means of molecular dynamics (MD) simulations for diamond and ta-C sliding contacts with monoatomic H and F surface passivation [18]. The latter work also shows that the corrugation of the CPES can be directly related to the steric hindrance between surface terminations (i.e., Pauli repulsion) that causes resistance to the sliding motion, at least for systems in which electrostatic interactions are not dominant. Given a tribological interface and a sliding direction, this observation allows to qualitatively rank different surface chemical terminations according to the interface shear stress by simply comparing the atomic-scale geometric features of the unloaded surfaces [18]. This is possible because the geometric structure and atomic-scale roughness of carbon surfaces with chemically stable passivation by monoatomic or diatomic species such as H, O and OH is hardly altered by the tribological load.

However, deriving such qualitative structure–property relationships is far more challenging when the surface dangling bonds are passivated by short chemisorbed molecular fragments, whose flexibility causes their geometric structure to change significantly upon tribological load. This is especially true when they are not closely packed on the surface [20]. This situation is common when organic friction modifiers are physisorbed or chemisorbed on steel surfaces [20,21,22]. It can also occur for diamond-like carbon surfaces as a result of the incomplete tribochemical degradation of the friction modifiers [7]—with chemisorption of lubricant molecules which can result in relatively dense surface layers [23,24]—or of tribologically induced surface oligomerization [16]. 

Since insights into the dynamic evolution of the atomic-scale structure of these chain-like surface terminations at tribological interfaces is virtually impossible experimentally, MD simulations have become an established tool for the investigation of the relationship between friction and the dynamic properties of chain-like surface terminations. For instance, MD studies with this goal were carried out on dry interfaces between self-assembled monolayers chemisorbed on silica [25] and gold [26] surfaces, as well as on lubricated interfaces between organic friction modifiers and surfactants that are physisorbed on iron oxide [20,22,27]. In these studies, the dependency of friction on parameters such as length, conformation, disorder, head group and surface coverage of the adsorbed chains as well as on external parameters such as normal pressure, sliding speed and temperature were investigated. The three aforementioned studies on boundary lubricated interfaces [20,22,27] highlight an aspect that hints at a similar structure-property relation as in the case of monoatomic passivations: friction is significantly influenced by the interdigitation between the friction modifier surface films and the lubricant molecules. The interdigitation is influenced by the chain surface coverage, their conformation and their head group. In general, friction tends to decrease at increasing coverages because dense surface molecular packing limits interdigitation, thus favouring the formation of a well-defined, solid-like slip plane between adsorbed films and lubricant. This effect was not investigated on the dry interfaces between self-assembled monolayers on silica and gold [25,26] owing to their very high surface coverage leading to a solid-like behavior of the interface. Moreover, early MD studies on the friction of diamond and a-C interfaces with chemisorbed hydrocarbon fragments [28,29] did not provide useful information on this aspect because the simulations were carried out on interfaces in which only one of the two surfaces was terminated with chain-like molecules, while the countersurface was hydrogen-terminated. 

The main goal of the present work is to complement the previously mentioned MD studies and, in particular, to gain further atomic-scale insights into the relationship between interface shear stress and surface chemical passivation for dry interfaces between a-C surfaces that are decorated by chemisorbed molecules. To do so, we performed a non-reactive, classical MD simulation study of the sliding friction between two a-C coatings decorated by different organic chain-like molecules that are chemisorbed onto the contacting surfaces under a normal pressure of 1 GPa and a sliding velocity of 2 m/s. We investigated the influence of the length, grafting density and polarity of the chains by studying almost 100 model tribological systems with different numbers of linear alkanes or alcohols of varying length attached to the a-C surfaces, also including different a-C densities. Finally, tribological contacts between the purely H-passivated a-C surfaces were also included in the parameter set as reference systems. 

Our analysis has three primary objectives that were not explicitly covered by previous studies [20,22,25,26,27]. The first objective is to provide a direct comparison between monoatomic and chain-like surface passivation and to understand whether a classification of the shear stress according to the properties of the unloaded systems (i.e., a-C density, length, density and polarity of the surface passivation species) can provide at least qualitative conclusions about the relationship between friction and surface passivation as it does in the case of monoatomic surface passivation [18]. The second objective is to extend the findings of Reichenbach et al. [18] and Kuwahara et al. [16] on the correlation between shear stress and dynamic steric hindrance between surface terminations in carbon tribological contacts. In particular, we aim at investigating whether, under tribological load, it is possible to find a robust correlation between the shear stress and the dynamic steric interlocking between chemical passivation species based on average density profiles that are commonly used to estimate the interlocking (e.g., Reference [20]). The last objective is to devise a more precise, yet simple descriptor of the dynamic interdigitation of the passivation species during sliding that correlates with the interface shear stress. The descriptor is based on a simple geometric evaluation of the maximum overlap between atoms of the two contacting surfaces. In contrast with averaged density profiles normally used for this purpose, the overlap is measured so that it properly captures the interlocking of the two surfaces that causes resistance to the sliding motion.

## 2. Materials and Methods

### 2.1. Preparation of Model Systems

Bulk a-C model systems are obtained by constant-volume quenching from the liquid by means of reactive MD simulations using the screened version of the REBO2 interatomic potential [30]. Following a standard procedure [31], we initially distribute carbon atoms randomly in an orthorhombic cell with size 21.58×19.93×30.52 Å^3^ to obtain systems with densities of 2.0, 2.6 and 3.2 g/cm^3^. After an equilibration step at 5000 K for 50 ps, we quench the system at constant volume to 300 K within 10 ps using a Langevin thermostat with time constant 0.5 ps. The equations of motion are integrated with a 0.1 fs time step.

The surfaces are obtained by releasing the periodic boundary conditions along the z direction (i.e., the direction perpendicular to the sliding plane). Surface carbon atoms left with only one first neighbor (the atomic coordination is calculated using a cutoff radius of 1.85 Å) are replaced by a hydrogen passivation atom. Otherwise, for each C–C bond that is cleaved by releasing the periodic boundary conditions, we passivate the corresponding dangling bond by adding a monovalent chemical species to the system. Dangling bonds on C atoms that were initially in an sp^3^ configuration are passivated with hydrogen atoms, while dangling bonds on C atoms that were initially in an sp^2^ configuration are passivated with H, OH, C_5_H_11_, C_5_H_10_OH, C_10_H_21_, or C_10_H_20_OH depending on the desired passivation structure.

### 2.2. Force Field

The interatomic potential used for the sliding MD simulations is a non-reactive force field with the analytic form and combination rules of the Optimized Potential for Liquid Simulations (OPLS) [32]. The force field parameters for bulk a-C and surface atoms are described in the following two subsections and the complete parameter set is provided in Table A1, Table A2, Table A3, Table A4 and Table A5 in Appendix A.

#### 2.2.1. Bulk Amorphous Carbon 

Following the approach by Reichenbach et al. [18], a-C is described using two atom types: carbon atoms in a diamond-like, sp^3^ configuration (atom type CD) and carbon atoms in a graphitic-like, sp^2^ configuration (atom type CG). After computing the neighbor lists with a cutoff of 1.85 Å, carbon atoms with 4 first neighbors are labelled as CD atoms while atoms with less than 4 neighbors are labelled as CG atoms. 

The bonded parameters for CD–CD two-body and CD-centered three-body terms are taken from Mayrhofer et al. [33] after rescaling the CD–CD bond length to obtain a diamond lattice constant of 3.567 Å [34]. The parameters can accurately describe the geometry and elastic constants of diamond [33]. CG–CG two-body terms and CG-centered three-body terms are described using the parameters for aromatic carbon atoms in 6-membered rings with one substituent that are defined in Amber [35], where the standard bonded parameters of OPLS are defined [32]. For the CG–CD two-body terms we use the arithmetic averages of the CD–CD and CG–CG force constants and equilibrium bond lengths. 

Finally, we introduce a short-range repulsion to mimic the Pauli repulsion between non-bonded CG atoms by truncating their Lennard-Jones interaction at its minimum (i.e., at a distance $2^{1/6}\sigma_{\mathrm{CG}}$), where their Lennard-Jones interaction energy is shifted to 0.0. No additional short-range repulsion is necessary for CD atoms because their four first neighbors prevent other CD/CG atoms from approaching. Following the OPLS conventions, we employ non-bonded interactions for atom pairs separated by at least three bonds. For atom pairs separated by exactly three bonds the non-bonded interaction is halved. In analogy to the bonding parameters, the Lennard-Jones parameters for CG are taken from the OPLS potential [32]. Our tests show that this simple choice of parameters provides an accurate description of the geometry and elastic properties of the a-C structures of interest for this study (Table 1).

#### 2.2.2. Surface Atoms

Surface atoms and their chemical passivation species (i.e., H atoms, OH groups, alkanes and alcohols) are described by standard OPLS parameters for hydrocarbon and alcohols [32]. For the harmonic bonds between surface C atoms and CG/CD atoms, we also classify surface C atoms into diamond-like and graphitic C atoms based on their number of first neighbors and use the same rules developed for the CD/CG interactions in the bulk.

### 2.3. Sliding Simulations

A typical model system used in the MD sliding simulations is shown in Figure 1a together with a schematic representation of the simulation parameters and of the constrained and thermostated regions. A normal load of 1 GPa and sliding velocity of 2 m/s along the x direction are imposed by external constraints on two rigid layers, one at the bottom and one at the top boundary of the tribological model system (Figure 1a). The normal load is controlled with the pressure-coupling algorithm described in Ref. [37]. The temperature is controlled by a Langevin thermostat with a target temperature of 300 K coupled to the $v_{y}$ velocity components of atoms inside the adjoining thermo regions (Figure 1a). The thickness in z direction of the rigid layers is 2 Å each, and of the thermostat layers 4.5 Å each. Periodic boundary conditions apply along the x and y directions. The time integration is performed with the software LAMMPS [38] using a timestep of 0.5 fs. After an equilibration period to reach a stationary state, the friction force $F_{\mathrm{shear}}\left(t\right)=\underset{i,j}{\sum }f_{x}^{i,j}\left(t\right)$ is extracted from the sum of all force components $f_{x}$ on atoms i in the upper sliding body caused by pairwise interactions with atoms j in the lower body. The average shear stress $P_{\mathrm{shear}}=\langle F_{\mathrm{shear}}\left(t\right)\rangle /\left(l_{x}l_{y}\right)$ is calculated from a time average over 15 ns (with $l_{x}$= 21.58 Å and $l_{y}$= 19.93 Å denoting the lengths of the simulation cell in x and y dimensions, i.e., $l_{x}l_{y}$ is the surface area). Snapshots of the interface regions after sliding for all considered alkane chain terminations on the a-C with the highest density are shown in Figure 1b. Details about the number of chain fragments per surface area for all considered a-C densities are summarized in Table 2.

## 3. Results

The shear stress results for all considered model sliding systems are summarized in Figure 2. The upper and lower plots show the results for non-polar and polar passivation species, respectively. As explained in the Section 2, each surface is passivated by a mixture of H atoms and chemisorbed rests R, with R defined in the legends on the right side of Figure 2. Since for R = H the surfaces are fully H-passivated independently of the number of R, only one system exists per a-C density. The results are still shown three times (white bars in the upper plot) to facilitate the comparison with the R = OH case (i.e., a mixture of H atoms and OH groups) in which the number of OH groups varies. Overall, we observe that the presence of the OH head group on the chains generally leads to higher shear stress $P_{\mathrm{shear}}$ compared to the non-polar systems. This is clearest for the short R species: for R = H, $P_{\mathrm{shear}}$ is always lower than for R = OH. Furthermore, the passivation of the a-C surfaces with single H atoms (R = H) or a mixture of H atoms and OH groups (R = OH) always results in a lower $P_{\mathrm{shear}}$ compared to the chain-passivated counterparts (chain length C_5_ or longer, see legend in Figure 2).

While an influence of the OH head groups on the shear stress can be observed, at least qualitatively, the influence of the other parameters is less obvious from this graphical presentation. Focusing on the non-polar systems, we see a systematic decrease in $P_{\mathrm{shear}}$ with increasing a-C density for the R = H case, which could be caused by a decreasing geometric corrugation of the a-C surfaces. However, with chemisorbed chains there is no clear correlation between the a-C density and the shear stress. In other words, the presence of the chain-like terminations fully masks the effect of the underlying a-C in terms of friction behavior. The effect of the different chain lengths (different colors in Figure 2) and number of chemisorbed chains (color intensity) on the shear stress is also not obvious from this plot. However, we note that the smallest $P_{\mathrm{shear}}$ values for chain systems are generally obtained for high densities of chains with equal length. From the snapshots in Figure 1, we can see that this combination leads to a dense “carpet” of straight perpendicular chains, which builds a relatively smooth surface and well defined, solid-like sliding interface. In contrast, for lower chain densities or mixtures of different chain lengths, we observe rougher surfaces and even clear interdigitation between chains on the lower and upper surface. Clearly, in the former case, in which chains form a dense, smooth carpet a lower shear stress can be expected than in the second case. 

The correlation between chain interdigitation and friction can be further evaluated based on the profiles along the system height z for the mass density ρ and velocity component in sliding direction $v_{x}$. These density profiles ρ(z) and velocity profiles $v_{x}\left(z\right)$ are obtained separately for the lower and upper body, as defined in Figure 3a on the example of the densest H-passivated a-C system, from time averaging 5 ns of the MD trajectories during sliding. Figure 3b,c show the profiles for the systems with alkane chains. The gray shaded areas mark the region where both lower (green) and upper (violet) density profiles are non-zero (note the logarithmic scale for the density, which strongly emphasizes the “overlap” of the profiles). The influence of this overlap can also be seen in the velocity profiles, where the step profile of a solid-like contact smoothens to a wider transition zone with a more continuous velocity gradient. As a comparison, we note that the H-passivated, dense a-C system in Figure 3a builds a very sharp interface without overlap of the density profiles and a step-like velocity profile. Figure 3d shows the shear stress results as a function of the density profile overlap for all considered systems (all a-C densities, all chain lengths as well as both alkane and alcohol). As expected, the shear stress tends to be higher in systems with larger overlap. Interestingly, for the reference systems (H-passivated a-C surfaces, full black circles in Figure 3d) the relation between this simple overlap parameter based on the density profiles and the shear stress is almost linear. However, for the chain-passivated systems it does not adequately explain the behavior of the shear force. Especially for the systems with mixed chain lengths, there is no particularly significant correlation. 

The reason why this simple overlap parameter correlates poorly with the shear stress can be seen when looking at the three-dimensional structure of the chains on the surface instead of considering the ρ(z) profile, which averages over the lateral dimensions x and y. An example can be seen in Figure 4, which shows a snapshot of the interface of the system with five decane molecules on each surface after sliding. Note that the empty space at the interface is just added in the graphical representation for a better visibility of the surfaces. The chains are very unevenly distributed on the surfaces, with areas of thicker coverage and areas of bare a-C surface on both sides, which match in a commensurable way. This leads to the large overlap of the density profiles observed in Figure 3. However, a large overlap between the density profiles of the two surfaces does not necessarily translate into a large steric interlocking in the direction of the sliding motion. Indeed, the roughness of the interface is anisotropic: the largest height variations of the order of 1 nm are observed perpendicular to the sliding direction, while the roughness in sliding direction is more on the scale of a single atom. This is likely due to a “streamlined” orientation of the chains in the sliding direction [20]. 

Hence, to find a more realistic descriptor of the steric hindrance to sliding, we divide the simulation box in *N* thin slices along the sliding direction (with *N* = 10, see Figure 4) and evaluate the following parameter within each slice *i*: the highest/lowest positioned surface atoms of the lower/upper body are identified and the height difference
(1)$$\Delta z^{\left(i\right)}\left(t\right)=z_{\min}^{\left(\mathrm{top},i\right)}\left(t\right)-z_{\max}^{\left(\mathrm{bottom},\;i\right)}\left(t\right)$$
is calculated. This calculation is performed dynamically during the sliding simulation within each slice. Finally, we reduce the result to a single value by averaging first over the sampling time τ and then over the *N* slabs. This parameter
(2)$$\bar{\Delta z}=\frac{1}{N\tau }\sum_{i=1}^{N}\int_{0}^{\tau }\Delta z^{\left(i\right)}\left(t\right)\;dt$$
is thus an estimate for the geometric overlap which the system encounters locally during sliding. For a slice width of about 0.2 nm (*N* = 10) the parameter $\bar{\Delta z}$ performs very well in qualitatively explaining the variation in the shear stress for the differently passivated surfaces, especially for the non-polar systems, as can be seen in Figure 5a (qualitatively compatible results were obtained using a similar approach by Kuwahara et al., on hydrogenated a-C contacts [16]). The results for the different chain types (squares, diamonds, triangles), as well as the solid a-C-surfaces without chains (circles), all fall on a single master curve. The shear stress increases linearly with decreasing $\bar{\Delta z}$, i.e., with decreasing steric hindrance. Note that the atoms’ center of mass positions are used for this evaluation, i.e., the size of the atoms is not taken into consideration. Typically, surface atoms are H atoms, which can be considered “in contact” at about the distance of their typical Lennard-Jones diameter $\sigma_{H}\sim 0.2$ nm (see Table A5). For smaller distances they have to overcome an energy barrier to pass each other during sliding. For larger values the interaction is weak, which can be interpreted as a “gap” between the surfaces. This reasoning is in good agreement with the offset seen in Figure 5. 

Moreover, we also observe a systematic correlation of $\bar{\Delta z}$ and $P_{\mathrm{shear}}$ with the chain properties, more precisely the surface density of chains. For the systems with the highest grafting density (red), the shear stress results are directly in line with the purely H-passivated a-C surfaces. Indeed, the surface structure formed by the –CH_3_ end groups of the close-packed chains (see Figure 1) looks very similar to the a-C-H surfaces and exhibits a similar “solid-like” behavior in the velocity profiles (Figure 3c). With decreasing number of chains (from red to blue in Figure 5a) $\bar{\Delta z}$ and $P_{\mathrm{shear}}$ increase. In parallel, the scattering of the data also increases. The latter might be explained by a larger chain mobility and configurational variety for loosely packed chains, which causes the emergence of repulsive forces of entropic origin as well as of other energy dissipation modes such as in polymer brushes systems [39]. As might be expected, the solid-like state is never reached in case of a passivation with chains of different length. In fact, we notice that the systems with mixed chain lengths are better characterized with respect to the influence of the chains density when only the number of longer chains is taken into account. This indicates that the steric interactions between the surfaces is mostly caused by the longer decane chains, which is obvious when the chains are oriented mostly perpendicular to the wall. In cases with very few chains, however, they tend to lay flat on the surface, which is likely another reason why the correlation between shear stress, local overlap parameter and chain grafting density are not as distinct for systems with low number of chains as they are for systems with higher density of chains. 

Finally, Figure 5b shows the same analysis for the systems with −OH head groups. Here, the correlation between $\bar{\Delta z}$ and $P_{\mathrm{shear}}$ is less significant compared to the alcane chains case. This suggests that the influence of the polarity is not well captured by the purely geometric analysis of the surface overlap. While the values for the local overlap parameter are not noticeably different, the shear stress is higher compared to the non-polar systems. The largest effect can be seen for densely packed chains of equal length (red squares and red diamonds). In these systems, the chains are oriented perpendicularly to the surface such that the chain heads with the polar group are located at the sliding interface. In contrast, for fewer chains or different chain lengths, some polar end groups are likely covered by other chains such that only part of the total number of −OH groups can actually contribute to an increased interaction between the two surfaces. In order to test these conclusions about the influence of the polar groups, the simulations and evaluations are repeated with “artificial” alcohol chains. These artificial chains are structurally identical to the alcohol chains, but with adapted charges on the −OH functionalized groups to suppress the polarity: the oxygen charge is set to zero, and the charges on the neighboring hydrogen and carbon atoms are set to the same value as if they were directly bonded to each other instead of via the oxygen atom (i.e., with definitions in Table A1: for the charges H4 is treated as H1, C6 as C5, and H5 as H2, CY as CZ; Lennard-Jones parameters remain unchanged). In this way, a comparison of the shear stress in these artificial test systems (Figure 5c) to the normal alcohol chain systems (Figure 5b) allows to separate the influence of the polar interactions from the effect of increased steric hindrance due to the presence of −OH groups. Indeed, the scatter of the shear stress results as a function of the overlap parameter is considerably smaller for the non-polar alcohol chains and is similar to the scatter of the alkane chain results. This confirms that polarity has an extra influence on friction which is not taken into consideration with the geometric steric hindrance parameter. 

## 4. Discussion

The main outcome of this work is the development of a simple, yet robust parameter that provides an estimate of the interdigitation of the chemical terminations of two a-C surfaces in relative motion in a dry tribological contact. The steric hindrance parameter is not as complex as other measures of the effective contact area between generic surfaces, such as the smooth-particle-based method developed and tested by Eder et al. [40]. However, its evaluation is straightforward and, unlike estimates of the interdigitation based on average density profiles [20], it allows the identification of a good qualitative correlation between steric resistance to sliding and interfacial shear stress. While this correlation only holds for systems in which the role of electrostatic interactions (polar heads) and entropic forces (long chains with rich configurational space) are not dominant, its evaluation provides a criterium to discriminate between friction mechanisms that depend on steric and non-steric effects. Due to its robustness, the interdigitation parameter lends itself to the analysis of a variety of tribological systems, whose surface chemical passivation ranges from monoatomic terminations (similar methodologies were used in Refs. [16,18]) to chain-like terminations, irrespective of the specific substrate. Future studies should aim at understanding the effects of sliding speed, pressure and temperature on the interdigitation, as well as the role played by the presence of residual lubricant in the contact (for this purpose a small modification of the steric hindrance parameter to account for the lubricant interdigitation should be developed). Furthermore, it would be interesting to investigate the consequences of the substitution of CH groups with CF groups. This is motivated by the radically different steric properties of CH and CF surface passivation and the resulting different frictional properties, both in terms of atomic surface passivation [18] and PTFE-like chain passivation [41].

The insights into the effect of specific chemical parameters on friction provided by our analysis are in agreement with the results of previous computational studies on different substrates. First, on average, the shear stress decreases with increasing surface density of termination species, as suggested by Ewen et al. [20] for iron oxide surfaces with adsorbed organic friction modifiers in lubricated contacts. Other studies on densely packed chain carpets confirm that densely packed chains lead to the formation of solid-like sliding interfaces [25,26,29]. Second, for such systems with densely packed chains, the shear stress becomes approximately independent of the chain length and of the properties of the substrate, as also observed by Chandross et al. for self-assembled monolayers on silica surfaces [25]. Finally, polar chain heads introduce further complexity that cannot be simply described by an evaluation of the geometric interlocking. In general, polar chain heads form hydrogen bonds with polar terminations on the opposite surface that need to be broken and reformed during sliding, thus leading to an enhanced corrugation of the energy landscape [42]. Moreover, for H/OH-terminated surfaces the geometric corrugation of the sliding contact is higher than for H-terminated surfaces, as observed in Ref. [3].

In quantitative terms, assuming an extended Amontons–Coulomb behavior of the shear stress ($P_{\mathrm{shear}}=P_{0}+\mu P_{z}$) with very small load-independent offset $P_{0}$ as found in previous studies for both monoatomically and chain-like passivated systems [11,18,25,26], the shear stress values resulting from our MD simulations of chain-passivated interfaces at $P_{z}=$ 1 GPa correspond to friction coefficients μ ranging from about 0.05 to about 0.45. Friction coefficients calculated in similar MD studies of chain-passivated iron oxide and silica surfaces [20,22,25,43] fall in the same range (~0.03–0.4). The size of this range of values depends on different aspects of the model system, such as the chain surface density, the presence or absence of residual lubricant molecules between the chain-terminated surfaces, the sliding speed or shear rate (μ is found to increase at increasing shear rates [20,26]). Experimental values of friction coefficients measured in boundary lubrication conditions on tribological systems with lubricant molecules adsorbed on the surfaces are typically close to 0.1 [21,24,43,44]. This roughly corresponds to the shear stress we found for non-polar systems with a relatively high chain surface density (about 5 nm^−2^). As shown in Figure 5a, higher shear stress values are caused by the increased interlocking of less dense chain interfaces. There might be several reasons why some of our systems show friction coefficients that are larger than typical experimental values. First, the chain surface density in the experiments could be higher than in some of our (higher friction) systems. Second, the experimental shear rate is typically at least two orders of magnitude lower than the simulated one, thus leading to lower friction coefficients in the experiments [20,26]. Finally, a direct quantitative comparison between friction coefficients in MD simulations and experiments is hardly possible, as mentioned in the Section 1, owing to the absence of a macroscopic surface roughness in the simulations.

The similarity between the friction mechanisms and shear stresses presented here for a-C surfaces and those found in other simulations works on different substrates (e.g., iron oxide [20,22,43], silicon oxide [25] and gold [26]) demonstrate the generality of these studies on chain-like boundary films. As shown in Figure 2, the presence of stable chemisorbed chain-like surface terminations reduces the importance of the substrate properties for friction. In both monoatomically and chain-like terminated tribological contacts the contact area and the local contact pressures are determined by the substrate’s mechanical properties and by its surface roughness [45]. Moreover, in the case of monoatomically surface terminations the contact shear stress is a direct consequence on the substrate stiffness and its atomic-scale surface corrugation [18]. Conversely, the main influence of the substrate on the contact shear stress of chain-terminated surfaces is related to its chemical interactions with the chemisorbed chains, which determines, at least in part, the chains surface density and their structure. 

Regarding the specific a-C surfaces investigated in this study, which play a major role in the friction properties of both crystalline and amorphous hard carbon coatings, as well as in the friction properties of surfaces which develop a carbon-rich surface film during running-in [46], we can state that in general surface passivation with H atoms or combinations of H and OH groups perform better than chain-like passivation in terms of shear stress. This is in agreement with the observation that superlow friction is usually achieved for atomically smooth, rigid and incommensurate contacts [19]. In this context, it is worthwhile noting that a dense packing of molecular fragments on such surfaces is difficult to achieve owing to the steric hindrance caused by preexisting surface chains that prevent other friction modifier molecules to reach reactive surface sites. In agreement with previous studies on superlubricity of diamond-like carbon coatings [7], a running-in phase with relatively high contact pressures should be helpful to achieve a full fragmentation of the lubricant molecules and thus a superlow friction state. Nevertheless, future MD studies considering surface roughness would be interesting to understand whether chain-passivation of surface regions outside the top of the asperities can be useful to mitigate the effect of roughness on friction.

## 5. Conclusions

Based on the three main objectives of this study as listed in the Section 1, our results can be summarized as follows:We find that a classification of the shear stress according to the properties of the unloaded systems (i.e., a-C density, length, density and polarity of the surface passivation species) only provides a few qualitative conclusions about the relationship between friction and surface passivation. Namely, chain-like passivation of the a-C surfaces leads to shear stresses that are consistently higher than those exhibited by H and H/OH-terminated systems but that do not depend on the density of the a-C substrate. Moreover, surface passivation species with OH head groups lead in general to higher friction than their non-polar counterparts. Finally, sliding systems with a high density of chains with equal length are those performing best among all considered chain-passivated systems in terms of shear stress.More precise insights into the friction–passivation relationship can be obtained through a careful analysis of the dynamic behavior of the sliding interfaces. A classification of the shear stress values according to the overlap between averaged density profiles of the passivation species delivers an overall qualitative trend whereby the shear stress increases with increasing overlap. However, the relation between this simple overlap parameter and the shear stress is almost linear only for monoatomically H-passivated a-C. This is because the overlap between averaged density profiles fails to properly describe the interlocking between surface chain-like passivation species along the sliding direction.Finally, we devise an improved descriptor of the interdigitation of the passivation species during sliding that can be well correlated with the shear stress. The descriptor is based on a simple geometric evaluation of the maximum overlap between atoms of the two contacting surfaces. In contrast with averaged density profiles, the overlap is measured so that it properly captures the interlocking of the two surfaces that causes resistance to the sliding motion. This descriptor of the steric hindrance explains most of the shear stress behavior also for systems with polar end groups (monohydric alcohol chains), but additional effects come into play owing to the electrostatic interactions between polar groups.

## Figures and Tables

**Figure 1 materials-15-03247-f001:**
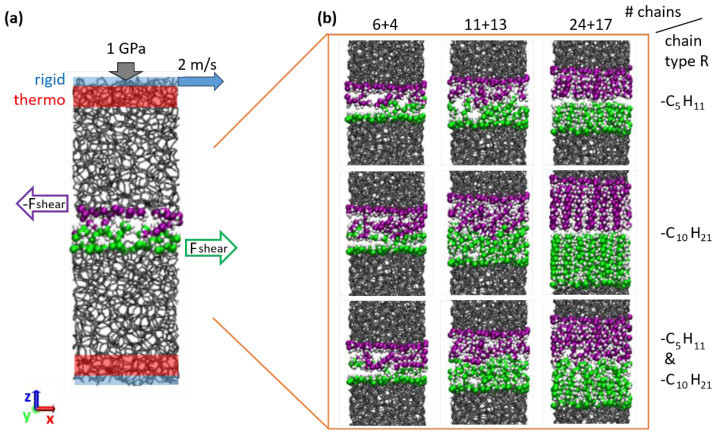
(**a**) Simulation setup for shear force calculation between two a-C blocks with alkane or alcohol chains chemisorbed on the surfaces in contact (gray sticks: bulk C-C bonds; spheres: surface atoms, white spheres represent H, green and violet spheres are C atoms of lower and upper surface, respectively). Blue shaded areas are rigid layers used to impose the normal load and the sliding velocity. The regions (red) adjacent to the rigid layers are coupled to a Langevin thermostat with a target temperature of 300 K. Further details are described in Materials and Methods. (**b**) Snapshots of the interface regions after sliding for all considered alkane chain terminations on the a-C with the highest density. Rows: different chain lengths (top to bottom: pentane; decane; mixture of both). Columns: different number (#) of grafted chains (increasing from left to right).

**Figure 2 materials-15-03247-f002:**
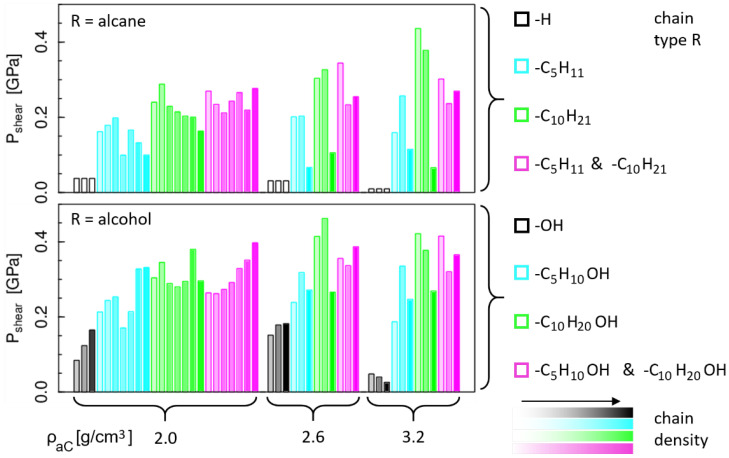
Summary of all shear stress results grouped by a-C density. Bar colors mark different chain types as listed in the legend, where the color intensity increases with the total number of grafted chains (i.e., chains on upper + lower surface). The upper and lower plots show results for non-polar and polar terminations, respectively. The three identical low $P_{\mathrm{shear}}$ results for each a-C density in the upper plot (white bars) refer to the same purely H-terminated system and are drawn three times to facilitate the comparison with the H/OH-terminated systems in which the number of OH groups varies (lower plot).

**Figure 3 materials-15-03247-f003:**
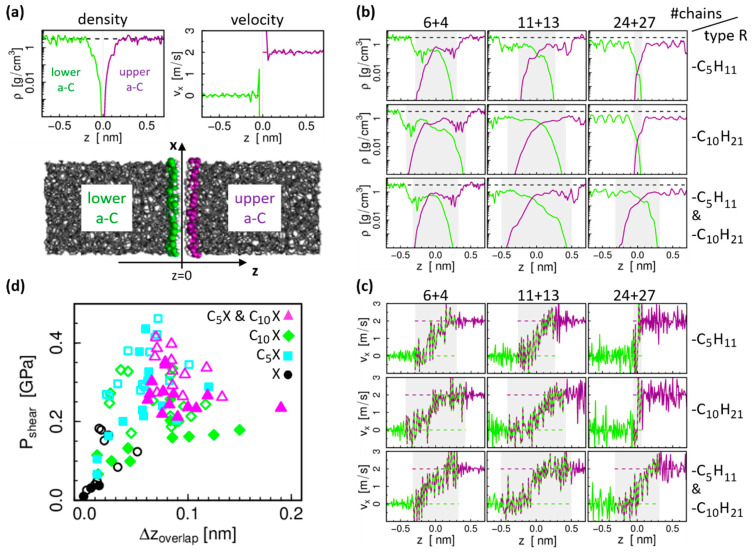
(**a**) Snapshot of the reference system without chains, i.e., purely H-passivated surfaces, and density and velocity profiles in the interface region; green: lower sliding body, violet: upper sliding body. (**b**,**c**) Density and velocity profiles for systems with chains (only highest a-C density); same color code used in (**a**); the presence of molecular fragments attached to the a-C surfaces leads to a considerable overlap of the lower/upper density profiles (gray shaded area) depending on the number (#) of chains. (**d**) Shear stress as a function of the interpenetration depth of the lower and upper body density profiles for all considered systems. Symbols mark different chain types (symbol shape refers to the chain length; full/open symbols refer to non-polar (X $\in \left\{H_{11};H_{21}\right\}$) or polar (X $\in \left\{H_{10}\mathrm{OH};H_{20}\mathrm{OH}\right\}$) chain types, respectively.

**Figure 4 materials-15-03247-f004:**
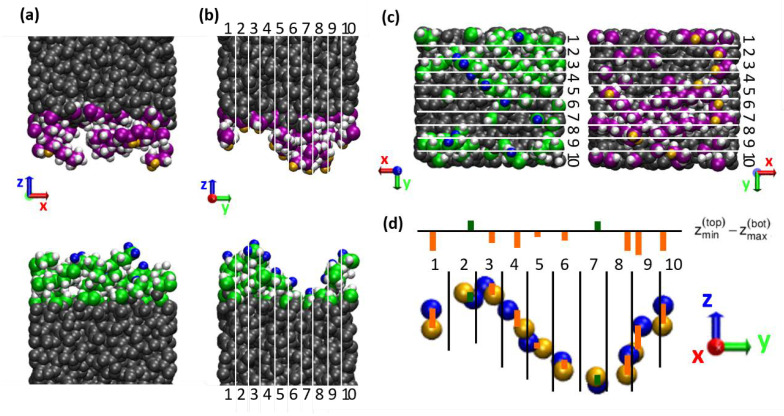
Two-dimensional evaluation of the surface interpenetration. (**a**–**c**) Snapshot of a system with 5 chains per surface seen from different perspectives: (**a**) perpendicular to and (**b**) along the sliding direction (red *x*-axis), as well as (**c**) top view on the lower/upper surface. Lower and upper body have been separated in the visualizations for better visibility of the surface atoms. The chains are unevenly distributed over the surface area, which leads to a high apparent interpenetration according to the 1D-density profiles (see Figure 3). The actual interpenetration of surface atoms felt during sliding is much smaller as only the roughness along the sliding direction is sampled. As a rough estimate for this effective roughness, the highest (blue) and lowest (yellow) positioned atoms of the lower and upper body, respectively, are identified in each stripe 1–10. (**d**) Difference between z-positions of the highest/lowest positioned surface atoms of lower/upper body per stripe (see Equation (1); orange/green = negative/positive Δz).

**Figure 5 materials-15-03247-f005:**
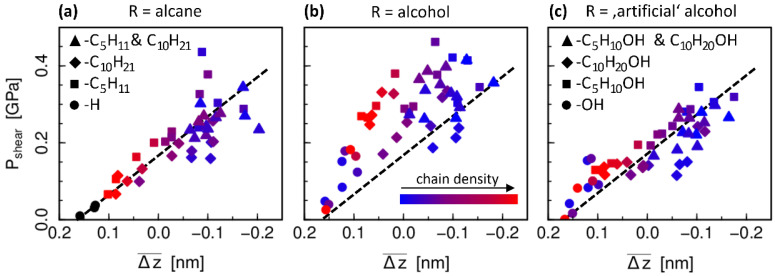
Shear stress as a function of the steric hindrance parameter $\bar{\Delta z}$ as defined in Equation (2) for all systems with (**a**) non-polar and (**b**) polar terminations, as well as (**c**) artificial alcohol terminations with adapted charges of the OH-functional groups (see text for further details). Symbols refer to different chain lengths (legend in (**c**) also holds for panel (**b**)) and the color code indicates the chain density (same in (**a**–**c**)), where only the longer chain type is taken into account here for the cases with mixed chain lengths (see text for further discussion). Black symbols in (**a**) refer to the fully H-terminated a-C systems. The dashed line is a guide to the eye for the slope of the correlation in (**a**) and is repeated in (**b**,**c**) for comparison.

**Table 1 materials-15-03247-t001:** Density, bulk modulus K, Young’s modulus E and Poisson’s ratio ν of the a-C samples used in this study as obtained by the presented force field and by the screened REBO2 potential (in brackets). $\rho_{0}$ refers to the density after the constant-volume quenching, $\rho_{\mathrm{relaxed}\;}$ is the density when the atomic positions and the simulation cell are relaxed to zero pressure. Additional reference data can be found in Ref. [36].

$\;\rho_{0}\;[g/{\mathrm{cm}}^{3}]$	$\rho_{\mathrm{relaxed}}\;[g/{\mathrm{cm}}^{3}]$	K	E	ν
2.0	2.22 (2.21)	243 (233)	337 (338)	0.27 (0.26)
2.6	2.67 (2.62)	380 (299)	589 (498)	0.24 (0.22)
3.2	3.27 (3.13)	467 (426)	993 (760)	0.15 (0.20)

**Table 2 materials-15-03247-t002:** List of all considered combinations of a-C-densities with different amounts of molecule fragments (“rests” R) chemisorbed on the lower and upper surfaces of the sliding contact. These combinations are studied for 8 different species R as defined in Figure 2: 6 chain types (3 alkane and 3 alcohol), -OH, and –H ^1^.

aC Density [g/cm^3^]	#Rests On Lower Surface	#Rests On Upper Surface	#Rests (Low + Up) Per Surface Area [nm^−2^]
3.2	24	27	11.86
	11	13	5.58
	6	4	2.32
2.6	29	22	11.86
	13	10	5.35
	6	5	2.56
2.0	24	21	10.46 ^2^
	20	19	9.07
	9	19	6.51 ^2^
	20	8	6.51 ^2^
	20	5	5.81 ^2^
	9	8	3.95
	5	5	2.32

^1^ For R = H, the surface is always fully H-terminated (i.e., the number of chains, #chains, is irrelevant) as explained in the text. ^2^ These combinations are only considered for the chain passivations (and not for R = OH or H).

## Data Availability

The data presented in this manuscript is available upon request from the authors.

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
