# Peer review of "Relating Dry Friction to Interdigitation of Surface Passivation Species: A Molecular Dynamics Study on Amorphous Carbon"

_materials, 2022, doi:10.3390/ma15093247_

Round 1

Reviewer 1 Report

In the manuscript materials-1618012, the authors studied the friction between dry amorphous carbon (a-C) surfaces with chemisorbed fragments of lubricant molecules employing molecular dynamic simulations. The motivation is timely, and the results are of the potential interest of Materials' readership. Moreover, the computational protocol was carefully described and employed in the simulations. In this sense, I recommend publishing the manuscript as it is.

Reviewer 2 Report

I think this is a well written and well presented article.  Dry amorphous carbon  surfaces with chemi-absorbed fragments of lubricant molecules is investigated employing molecular dynamic simulations for shear force variability.

I cannot object to its publication.

However, I do think that shear force and friction forces are being confused.  Molecular level force calculations are not the correct scale of analysis for friction even for very smooth surfaces.

The MD simulation calculates a shear force albeit under a normal load.

Friction is an entirely different concept.  It particularly important to distinguish between the two properties for dry contacts.  Dry contacts have hills and valleys.  

The article sometimes reads as if a physical experiment has been performed e.g. "The setup and parameters for the sliding MD simulations are summarized in Figure 1a. A normal load of 1 GPa and sliding velocity of 2 m/s along the ? direction are imposed by external constraints on two rigid layers, one at the bottom and one at the top boundary of the tribological system".

It is a simulation not a physical experiment so perhaps the authors can be a bit more clear about this aspect.  But in this filed it seems like several authors have used similar terminology.

What is important to consider are surface roughness, surface asperities etc. for dry friction.  Clearly the authors know this, as can be read in their last paragraph.

I cannot judge the MD work as there is not much information provided on the number of simulations.

Reviewer 3 Report

The article titled “Relating dry friction to interdigitation of surface passivation 2 species: a molecular dynamics study on amorphous carbon.” Is very well written and the figures are very well presented and described. However, the following comments need to be considered to improve the article further:

  1. The introduction is too lengthy that needs to be shortened and be more focused on the related previous studies.
  2. The objective for the study is not clear and needs to be drawn at the end of the introduction based on the discussed previous studies.
  3. Yes, the figure captions very well describe the figures but if slightly reduced in terms of description and give the detailed descriptions within the text, this might be better and make them look simpler.
  4. It is suggested to include the discussion section within the results section and make them one section Results and discussion
  5. A separate section for the conclusions which represent the main outcomes of the work in concise bullet points is essentially important to complete the structure of the article.

Reviewer 4 Report

The paper presents MD simulations of dry friction between monolayer boundary films on passivated amorphous diamond coatings. The simulation model, method and results are interesting. A few points should be clarified in revision.

1) The authors attribute the dry friction to the interdigitation of chemical terminations, and propose a steric hindrance parameter to characterize the interdigitation strength. All simulations were done under the same pressure of 1.0GPa, and the calculated shear stresses are in the range of 0.05--0.45 GPa, meaning that COF is in the range of 0.05--0.45 at the pressure of 1.0GPa, see Fig.2. Did the authors confirm that the MD simulations of friction yields Amontons-Coulomb law? If so, why is the calculated COF for the cases of C5H11, C10H21, C5H11 &-C10H21, C5H10OH,C10H20OH, and C5H10OH &-C10H20OH so high? In most of boundary lubrication tests, the measured COF value is around 0.1.

2) In the discussion part, quantitative comparisons with experiment measurements should be provided in revision.

3) What is the primary role of the substrate, crystalline or amorphous diamond? If the substrate is changed to oxides or bare metals, what will be mostly different in friction?

4) Conclusions are absent.
